# An Assessment of the Knowledge and Perceptions of Precision Medicine (PM) in the Rwandan Healthcare Setting

**DOI:** 10.3390/jpm13121707

**Published:** 2023-12-14

**Authors:** Clarisse Musanabaganwa, Hinda Ruton, Deogratias Ruhangaza, Nicaise Nsabimana, Emmanuel Kayitare, Thierry Zawadi Muvunyi, Muhammed Semakula, Faustin Ntirenganya, Emile Musoni, Jules Ndoli, Elisee Hategekimana, Angus Nassir, Francis Makokha, Aline Uwimana, Joel Gasana, Pierre Celestin Munezero, Francois Uwinkindi, Claude Mambo Muvunyi, Laetitia Nyirazinyoye, Jean Baptiste Mazarati, Leon Mutesa

**Affiliations:** 1Division of Research Innovation and Data Science, Rwanda Biomedical Center, Kigali P.O. Box 7162, Rwanda; semakulam@gmail.com (M.S.); joelgasana6@gmail.com (J.G.); claude.muvunyi@rbc.gov.rw (C.M.M.); 2Center of Human Genetics, College of Medicine and Health Sciences, University of Rwanda, Kigali P.O. Box 4285, Rwanda; 3School of Public Health, University of Rwanda, Kigali P.O. Box 3286, Rwanda; rutonh@gmail.com (H.R.); lnyirazi@gmail.com (L.N.); 4Butaro District Hospital, Burera P.O. Box 59, Rwanda; druhangaza@ughe.org (D.R.); nicaisensabimana@gmail.com (N.N.); kayitaree@gmail.com (E.K.); 5King Faisal Hospital, Kigali P.O. Box 2534, Rwanda; zathierry@yahoo.fr; 6University Teaching Hospital of Kigali, Kigali P.O. Box 655, Rwanda; fostino21@yahoo.fr (F.N.); musemile1@gmail.com (E.M.); 7University Teaching Hospital of Butare, Huye P.O. Box 254, Rwanda; jndoli1971@gmail.com (J.N.); hatel2020@gmail.com (E.H.); 8Kenya Institute of Bioinfomatics, Nairobi P.O. Box 918, Kenya; angusnassir2000@yahoo.com; 9Directorate of Research and Development, Mount Kenya University, Thika P.O. Box 342-01000, Kenya; makokhafw@gmail.com; 10Malaria and Other Parasitic Diseases Division, Rwanda Biomedical Center, Kigali P.O. Box 7162, Rwanda; aline.uwimana2015@gmail.com; 11Department of Microbiology and Parasitology, School of Medicine and Pharmacy, College of Medicine and Health Sciences, University of Rwanda, Huye P.O. Box 117, Rwanda; munezeropierrecelestin@gmail.com; 12Division of Non-Communicable Diseases, Rwanda Biomedical Center, Kigali P.O. Box 7162, Rwanda; uwifranco@gmail.com; 13School of Medicine, University of Global Health Equity, University of Global Health Equity, Kigali P.O. Box 6955, Rwanda; jmazarati@gmail.com

**Keywords:** precision or personalized medicine, breast cancer, healthcare providers, national policies, cancer healthcare services

## Abstract

Introduction: Precision medicine (PM) or personalized medicine is an innovative approach that aims to tailor disease prevention and treatment to consider the differences in people’s genes, environments, and lifestyles. Although many efforts have been made to accelerate the universal adoption of PM, several challenges need to be addressed in order to advance PM in Africa. Therefore, our study aimed to establish baseline data on the knowledge and perceptions of the implementation of PM in the Rwandan healthcare setting. Method: A descriptive qualitative study was conducted in five hospitals offering diagnostics and oncology services to cancer patients in Rwanda. To understand the existing policies regarding PM implementation in the country, two additional institutions were surveyed: the Ministry of Health (MOH), which creates and sets policies for the overall vision of the health sector, and the Rwanda Biomedical Center (RBC), which coordinates the implementation of health sector policies in the country. The researchers conducted 32 key informant interviews and assessed the functionality of available PM equipment in the 5 selected health facilities. The data were thematically categorized and analyzed. Results: The study revealed that PM is perceived as a complex and expensive program by most health managers and health providers. The most cited challenges to implementing PM included the following: the lack of policies and guidelines; the lack of supportive infrastructures and limited suppliers of required equipment and laboratory consumables; financial constraints; cultural, behavioral, and religious beliefs; and limited trained, motivated, and specialized healthcare providers. Regarding access to health services for cancer treatment, patients with health insurance pay 10% of their medical costs, which is still too expensive for Rwandans. Conclusion: The study participants highlighted the importance of PM to enhance healthcare delivery if the identified barriers are addressed. For instance, Rwandan health sector leadership might consider the creation of specialized oncology centers in all or some referral hospitals with all the necessary genomic equipment and trained staff to serve the needs of the country and implement a PM program.

## 1. Introduction

In 2020, there were an estimated 19.3 million new cancer cases and 10.0 million cancer deaths across the world [1]. The most commonly diagnosed cancer was breast cancer, with an estimated incidence of 2.26 million new cases. Lung cancer was the leading cause of cancer deaths, with an estimated 1.8 million deaths [2]. Cancer research has led to new discoveries and sophisticated technologies, which enable the implementation of a cancer precision medicine (CPM) system [3]. The CPM system provides a wide range of options for cancer management, such as screening, the selection and prediction of effective drugs and treatments, the monitoring of relapse or recurrence, and customized immunotherapy [3,4,5]. In this system, individual cancer patients can expect to receive personalized care, with an appropriate dose of the right drug at the right time [6]. Precision medicine (PM) is an innovative approach that aims to tailor disease prevention and treatment by considering patients’ differences. On the other hand, as PM provides treatment options for patients with a particular genetic makeup, the cost of PM treatment is greater than the existing options [7]. One such example is trastuzumab, which was approved by the US Food and Drug Administration in 2006 for the treatment of ERBB2 (formerly HER2 or HER2/neu) overexpression in breast cancer. This therapy is not easily accessible in low and middle income countries (LMICs) due to the high cost associated with the treatment [8].

Studies have been conducted to evaluate the knowledge and awareness of PM in treating cancer. One study reported a very low familiarity with the term “PM” and a refusal to participate in clinical trials due to fear of receiving the placebo, which may affect the right course of treatment [9]. Although the PM field is advancing in high-income countries, there is still a low implementation of PM in LMICs due to a lack of funding and limited awareness and understanding of its potential impact [10]. Additionally, a study conducted in the African region revealed that over 80% of the study participants, comprising physicians, postgraduate students, laboratory heads, and scientists engaged in research activities, were using or planning to use advancing genomic technologies in their research. However, the study also highlighted the limited awareness of the PM field. Only 14% of the study participants reported that scientists in their countries are ready to implement PM or genomic medicine. Moreover, more than 40% reported needing more information and access to the appropriate infrastructure [11]. The unavailability of infrastructure and a qualified scientific capacity, limited resources, and suboptimal regulations are major barriers to the provision of PM in LMICs. All the above-mentioned challenges could worsen global health inequality, as patients in LMICs could die from preventable diseases. Therefore, it is crucial to establish and strengthen PM in LMIC healthcare settings.

Rwanda has made significant advances in enhancing healthcare delivery to its population. However, Rwanda needs to establish new innovative evidence-based interventions to respond to the growing cancer rates. In 2018, there were approximately 10,704 new cancer diagnoses in Rwanda [12]. The national strategies, such as the establishment of national cancer control plans and a national cancer registry for the collection, storage, and management of data on cancer cases, have enabled cancer surveillance in the country. Thus far, Rwanda has been progressing to equip healthcare facilities to carry out advanced molecular diagnostic techniques capable of generating genomic sequencing data that may be used to better manage and prevent cancer diseases. Moreover, Rwanda continues to promote science and technology through research and innovation. Technologies such as PM are among the current priorities for the country. However, there is limited information on the knowledge and perception of PM among healthcare professionals. Therefore, the aim of this research study was to assess the knowledge of healthcare providers and decision makers on PM and the feasibility of its implementation in the Rwandan healthcare setting.

## 2. Materials and Methods

### 2.1. Study Design

In June 2020–2021, a descriptive qualitative study was conducted using key informants’ interviews of hospital staff and direct observations of PM equipment functionality in selected Rwandan health facilities that host centers for the diagnosis and treatment of cancers.

### 2.2. Study Population

The study participants were staff working in the fields of oncology, pathology, genetics, or obstetrics working in the departments of oncology or providing cancer-related healthcare services at five health facilities: University Teaching Hospital of Kigali (CHUK); Butaro District Hospital (BDH); University Teaching Hospital of Butare (CHUB); King Faisal Hospital (KFH); and the Rwanda Military Hospital (RMH). Additionally, this study included health professionals from the Ministry of Health (MOH), which is a health sector policy-formulating entity, and Rwanda Biomedical Center (RBC), which is the implementing agency of health sector policies. The staff who were available during the data collection period (May 2020–November 2021) were interviewed.

### 2.3. Sampling Procedures

Purposive sampling was used to recruit study participants for interviews. The process consisted of continuous recruiting until theoretical saturation was reached [13]. Eligible participants were identified by the research team with the assistance of institutional leaders in the MOH, RBC, and the five health facilities. A representative of the research team worked closely in collaboration with the institutional leaders to recruit eligible participants and arrange interviews. The participants were contacted by email, phone calls, and face-to-face and online meetings. Informed consent forms were signed by the participants before each interview. For online interviews, signed consent forms were returned to the research team via email. The total number of participants from each site was five, with one representative from each of the following categories: histopathology, oncology, genetics, molecular biology, obstetrics, or surgery. An additional seven interviews were conducted with decision makers from the MOH and RBC.

### 2.4. Data Collection

Prior to data collection, 1 day of training for the research assistants and data coders was conducted by the research team leader. The training included a review of informed consent and data collection procedures and an introduction to PM and its implications for the healthcare system.

In-depth individual interview (IDI) questionnaires were adapted from an online African Precision Medicine survey. Open-ended questions on PM leadership, governance, financing, policies, and strategies were added to the questionnaire in order to understand the national strategic direction and progress toward PM implementation. Some IDI questions were closed-ended, and their answers were recorded using Likert scales. All interviews were audio-recorded using tablets and transcribed into Microsoft Word. A sample of transcripts was randomly and continuously selected to assess for possible translation errors. Errors were reviewed with transcribers to improve the quality of the transcripts.

Data managers and senior scientists in oncology services were asked to guide the research team to observe the available infrastructure and equipment that support PM implementation. Where possible, pictures of each item observed were taken and recorded. A seven-item checklist was used to collect additional information on availability and functionality.

### 2.5. Data Analysis

The data were cleaned and coded using Microsoft Excel to identify the main themes and subthemes of the participant responses. Under each subtheme, ideas were thematically analyzed and summarized, and relevant quotes from the participants were highlighted. Responses recorded using Likert scales were summarized using ranges and means. Observational field reports and pictures were analyzed manually by the team leaders, and descriptions of the equipment and technologies observed and their functionality were given.

### 2.6. Ethical Considerations

Ethical approval with the reference number 161/RNEC/2020 was obtained from the National Ethics Committee on 27 May 2020. Confidentiality was enhanced by using unique identifiers instead of names. Informed consent was obtained from the eligible participants and kept in a locked cabinet in the team leader’s office.

## 3. Results

### 3.1. Key Themes

#### 3.1.1. Perceptions of Participants about Knowledge of PM among Decision Makers and Breast Cancer Healthcare Providers in Rwanda

Among the 30 participants who responded to this question, 12 (40%) rated themselves as having adequate knowledge and being familiar with the concept of PM. Ten (33.3%) participants rated themselves as not being familiar and without enough knowledge, and 8 (26.7%) participants rated themselves as having moderate knowledge of the concept of PM (Figure 1). One respondent defined PM as the following:
“PM is about customization of treatment of a patient depending on her/his lifestyle or depending on his genetic make-up. There are some biomarkers that can be analyzed to orient or to customize a specific treatment depending on the genetic data of the patient”(Appendix A)

#### 3.1.2. Willingness of Institutions to Invest in a PM Program and the Perception of Individuals’ Plans to Use It in Their Work

Among the 25 participants who responded to this question, 12 of them indicated that they were not sure about their institution’s level of investment in PM programs. A total of 11 participants affirmed that their institutions are willing to invest in PM programs, while 2 stated that, to their knowledge, their respective institutions are not ready and not willing to invest in PM. This study also sought to explore more about the perceptions of individuals’ plans to use PM in their work. A total of 23 participants responded to this question, and among them, 19 confirmed that they are planning to use PM in their work, whereas 3 were not sure about it, and only 1 said they were not planning to use PM (Figure 2). One respondent highlighted their feelings as follows:
“Basing on the current situation with the available infrastructures and technologies, I think we can use PM in our settings especially in referral hospitals. The technology we have might facilitate the initiation of PM program… said by geneticist expert”(Appendix A)

#### 3.1.3. Perceived Barriers to Implementing PM programs in Rwandan Health Facilities

The participants were asked about the barriers to implementing PM in Rwanda. Among the 29 participants who responded to the questions, 27 affirmed that the lack of supporting infrastructure and technologies is the major barrier, followed by 25 who mentioned financial, cultural, and religious barriers as the issues hindering the implementation of PM, and 22 reported the limited education of PM as the barrier to implementing PM in Rwanda, whereas 16 reported insufficient information on the determinants of diseases as well as the high costs associated with the implementation of PM. A total of 13 participants said that the low priority given to PM was considered a barrier. Another barrier cited by 10 participants was the lack of ethical, legal, and social frameworks, and another 8 mentioned the fear of implementing PM (Figure 3). Tradition healers, extreme poverty, and copayments for patients using health insurance were perceived as barriers to PM use in Rwandan health facilities. Most of the participants perceived insurance coverage as a major barrier to the implementation of PM. One participant highlighted the following:
“Copayment (10%) for patients using health insurance: This copayment is not affordable for cancer patients and discourage them for seeking medical services especially those living in extreme poverty”(Appendix A)

#### 3.1.4. Integration of PM Programs into the Existing Technologies and Health System in Rwanda

All participants had a positive perception about the integration of PM into the existing technologies and health system in Rwanda. Most of them indicated that PM can be integrated well into the existing technology, as the Rwandan health system is well organized from teaching hospitals down at the community level. They also added that PM can benefit from our settings if the National Reference Laboratory (NRL) is able to perform different types of tests that feed the PM program. Healthcare providers from oncology services mentioned that PM is not new in Rwanda. In real practice, the participants affirmed that PM is less integrated in the current healthcare system, although some equipment can help with the implementation of PM programs, such as those at the NRL. Additionally, a feasibility study was suggested to be conducted in all healthcare facilities to identify the required technologies and human resources to be trained to drive the implementation of PM programs in their settings.

#### 3.1.5. Perceived Changes in Government Policies to Support the Implementation of PM in Rwanda

The interviewed health care providers did not know if there were policies in place to support the implementation of PM in Rwanda. Therefore, they stated that they did not perceive any changes. However, a number of healthcare providers involved in this research suggested that there is a need to update health sector policies and ensure that details on how biological materials from patients and other important data could be used in exchange programs with experts to facilitate PM implementation. The participants expressed their feelings, saying the following:
*“I don’t know if there is any specific policy on PM, I think the principles of PM are just incorporated in different guidelines, not specific guidelines for PM; so, the improvement can be done if we have people to do it, also capacity building is necessary, currently, we have some local experts and policy makers who can think about what changes are required to promote PM implementation in Rwandan health care settings.”*

The study participants suggested that awareness of the use of PM in the Rwandan healthcare setting could be a responsibility for all health professionals by putting efforts into understanding the PM concept. Not only could they understand PM, but they could also use it in their daily healthcare routine activities and eventually accelerate the implementation of this program. The study participants also suggested that the advocacy and mobilization of funds and institutionalizing PM legal frameworks (protocols, policy and procedure manuals, and guidelines) could be a priority to promote PM implementation in their country. For example, updating the current national health sector policy by adding key national strategies to accelerate the implementation of PM in their country was advised. Examples of the key strategies proposed include the following: the establishment of genomic surveillance programs in all health facilities, prioritizing cancer centers of excellence, as well as the development of legal frameworks to govern and regulate the process. The public–private partnership model (PPP) was another approach proposed to sustain national PM plans. Efforts to invest in education, focusing on health programs linked to PM, might also be a priority. For instance, the study participants suggested that the Rwandan medical education curriculum could be revised to include specific courses on PM. Moreover, it was suggested that the creation of new programs specializing in PM cancer management and the establishment of a multidisciplinary taskforce including medical specialists, senior scientists in medical science, data scientists, and bioinformaticians will contribute toward accelerating PM initiatives in Rwanda.

#### 3.1.6. Research and Training Needs for Healthcare Providers to Accelerate PM Program Implementation in Rwandan Settings

The study participants who were interviewed expressed that PM implementation requires skilled human resources. Therefore, training healthcare professionals through international academic exchange programs could be a starting point. A total of 20 out of the 25 study participants said that there is a need for more education surrounding PM. Twenty-one out of 23 participants said that it is necessary to train laboratory technicians. The training of genetic counsellors was suggested by 22 respondents, whereas 19 mentioned that it is necessary to create or expand a bachelor’s degree or MSc program in PM. The training of nurses and primary caregivers was also suggested by 19 out of 20 participants who responded to this question. Seventeen out of 19 respondents mentioned the necessity of sending students and physicians abroad for specialized training (Table 1).

#### 3.1.7. Perceptions about Patient Data Collection in a Rwandan Healthcare Setting

Information concerning patient details and documentation status in routine healthcare settings is important to know, as data are the main tools that guide PM (Figure 4). Almost all participants (96%) mentioned that patients’ data are stored in an electronic system, and 88% of the study participants responded that they share patient information with collaborators for clinical care purposes or research purposes upon approval by ethical review committees and the hospital leadership. Meanwhile, 92% reported routine collection of demographic information in five health facilities. The collection of genomic data in the health care facilities was reported by 23% of participants, and only 7% of the study participants reported that they do not collect patients’ data routinely. (These were the study participants from the MOH or RBC, as they do not meet patients in their routine activities).

#### 3.1.8. Access to the Required Infrastructure for PM in Rwandan Cancer Centers

The study participants thought that the existing technology in Rwanda was sufficient to implement PM. Others said that only basic technology is available in Rwandan healthcare settings. However, they insisted that the available technology was not enough to produce the expected PM outputs. They added that it may be possible to use the existing technology if additional advanced equipment, such as next-generation sequencing (NGS) machines and high computing technology, is installed in the health facilities, such as all referral and provincial hospitals. The health professionals in the RBC mentioned that there is already availability of the required advanced equipment, such as NGS platforms, at the NRL. However, that equipment is used for the genomic surveillance of HIV and TB. They suggested that if the institution is able to secure a stable supply chain of reagents, then the equipment could be used in other programs linked to PM.

Two cancer centers (CHUB and RMH) own biobank infrastructures for preserving patient samples for research purposes. They also have a separate clinical and research laboratory facility. Other centers like CHUK and BDH reported utilizing the available laboratory facility for both diagnostic and research activities. The available equipment in all the centers was in good condition and functioning well. Although not all centers have a comprehensive genomic surveillance facility, all centers demonstrated the ability to have basic computational facilities for data analysis, such as data storage and archiving facilities, including servers. Biobank spaces that can be used for research purpose were available in CHUK, RMH, and CHUB. Biosafety cabinets and PCR machines for research purposes were also available in CHUK and RMH.

## 4. Discussion

The individual’s genetic profiles are the basis of PM. The purpose of PM is to predict the most effective treatment for a patient. It has the potential to eliminate health disparities once it becomes widely available [14,15]. This study was carried out to assess the knowledge and awareness of healthcare providers of the implementation of the PM program in Rwandan settings.

### 4.1. Awareness and Perception of PM

The knowledge and familiarity with the concept of PM were moderate among the study participants. Although the study participants were unsure of the meaning of PM, a number of them defined it as a medicine tailored to an individual, and a few of them mentioned the molecular and environmental characteristics as crucial factors for the implementation of PM. On the other hand, study participants such as clinicians from oncology services and senior scientists from the fields of genetics and molecular biology were reportedly aware of the concept of PM and ready to use it. However, the findings of this study showed that there is still a need to raise awareness of PM, targeting healthcare providers working in other clinical care services. Although previous studies revealed the importance of genetic information, patients’ lifestyles, and environmental conditions as key drivers of PM [16], and Naithani et al. suggested that social, behavioral, and environmental factors are important on top of genetic information [17], the findings of this study show that the contribution of one’s lifestyle and environment in implementing a PM program was not commonly thought about by the study participants.

Not surprisingly, poor knowledge and low familiarity with PM among healthcare providers have been reported in different countries and regions of the world [18,19,20,21]. Knowledge and skills concerning the field of genetics are a requirement to ensure that health professionals deliver the best care when implementing PM. However, this study showed that there are gaps in human resources, especially in the field of genetics. The absence of genomic medicine experts was also highlighted in previous studies as a factor that delays the efforts to establish genomic technologies [22]. Expertise at a multidisciplinary level, including specialization in biology, bioinformatics, and clinical genetics, is required for an accurate interpretation of patient data and the use of PM in disease management [23]. Although the value of integration of PM programs in the healthcare system was recognized by the study participants, our findings showed that the perceived relevance of PM use was variable and linked to different institutional practices and standards of healthcare. These findings were coherent with other published data in South Africa, where the perceived relevance of PM to individual practice was also linked to varying practice standards across medical specialties [24]. Additionally, these findings support the points made by other scientists who advocate for governmental support to promote and fund PM in all aspects of implementation [25].

### 4.2. Limited Education

This study also explored PM training needs in Rwandan healthcare settings. The study participants suggested providing opportunities for training on PM concepts to key healthcare professionals such as laboratory technicians, genetic counsellors, nurses, and primary caregivers. Additionally, the establishment of a bachelor’s or master’s degree program in PM was recommended. The need to elaborate on these types of training was similarly pinpointed in studies conducted in South Africa on clinicians’ perceptions of PM as well as on the list of African PM needs [24,26]. Fortunately, there are ongoing initiatives for the education of Rwandan medical residents at CHUK concerning PM [27]. Hopefully, this will deal with some of the challenges reported in this study, especially concerning the awareness of PM and capacity building in Rwanda.

### 4.3. Research Infrastructure

The evidence-based process in the utilization of the PM concept in a healthcare system considers the importance of translating genomic data to clinical information for different purposes, including diagnosis, prognosis, therapy, and disease monitoring [28]. NGS technology has the capacity to sequence multiple genes at once and provide information on disease-associated variants, which are critical in matching patients to their appropriate therapies or uncovering disease risks [29]. Our study findings showed that there is a willingness for Rwandan institutions to invest in PM. However, this did not remove the barriers, such as limited supportive infrastructure and technology, including genomic facilities in hospitals. Moreover, although there was a positive perception of the integration of the PM program with the existing technology, the participants affirmed that the PM program is less integrated into the current healthcare system.

### 4.4. Patient Data

Regarding patient data collected in routine healthcare settings, this study revealed that Rwandan healthcare facilities meet acceptable standards for data management and archiving. Moreover, the participants reported that there is routine data collection of patient demographic and clinical information. However, there is still a gap in the collection of genetic or genomic information due to the limited genomic infrastructure in hospitals. The findings of this study are consistent with the existing literature, where limited clinical data on the determinants of diseases and limited availability of genetic data, as well as complex logistics, were reported as barriers hindering the integration of PM into cancer treatment [30,31,32]. In fact, Erdman et al. reported that a lack of sufficient genetic, environmental, and lifestyle information is a barrier to implementing PM [33]. Additionally, financial resources [34] and cultural, behavioral, and religious beliefs were also reported as barriers to PM. Yeary et al. highlighted religious beliefs as a factor to be considered in PM implementation because of its physical and mental health aspects for many people [35].

### 4.5. Ethics and Legal Framework

PM policy and other strategic documents were reported to be lacking and probably not yet integrated into the current national health sector policies. This is consistent with the data in the literature, where concerns with ethical, legal, and social frameworks and the fear of implementing PM remain as unresolved challenges in many countries [36]. Therefore, there is a need to establish legal frameworks that will guide the implementation of PM programs in Rwandan healthcare facilities. Most importantly, it is advised to consider PM programs in national health sector policy and strategic plans to ensure that the cost of PM activities is prioritized at the national level.

### 4.6. Limited Financial Resources

However, it is still difficult to accurately estimate the cost of precision medicine due to the inconsistencies in the methodology and the heterogeneity of the data. A scoping review demonstrated some key factors that influence the cost-effectiveness of precision medicine, such as the prevalence of genetic conditions in the group of targets, the cost of genetic testing and the treatment needed, and complication and death risk probabilities [37]. Because of the additional testing needed to improve stratification, PM interventions are highly costly compared with non-PM interventions. Although it is expected to decrease in the future, the high cost of genetic tests constitutes one of the barriers to the cost-effectiveness of PM intervention. Plöthner et al. demonstrated pharmacogenetic test-guided therapy as an opportunity to improve the cost-effectiveness of pharmacotherapy [38]. The genetic test is promoted with companion treatment, which is also highly costly. Saving costs in PM can also be affected by the accuracy of the genetic tests, owing to unnecessary therapeutic interventions or mortality or morbidity associated with false-positive or false-negative results, respectively [37].

### 4.7. Study Strengths and Limitations

The strength of the current study is that we interviewed key health professionals, decision makers, and scientists in all five Rwandan hospitals involved in cancer management (i.e., CHUK, CHUB, BDH, KFH, and RMH). To gain more insights into the perceptions and knowledge of PM implementation at the national level, this study also interviewed health professionals from the RBC, a health sector implementing entity that promotes high-quality, affordable, and sustainable healthcare services to the population through evidence-based interventions. Moreover, policymakers from the MOH were interviewed, as it is an institution that establishes all health sector policies on behalf of the government of Rwanda. There were some limitations in this study, such as the small sample sizes for each represented field. This was due to the limited number of specialists and experts in fields such as genetics and molecular biology. Therefore, different studies are suggested to extend to other neighboring countries that share the same problem in order to achieve a clear picture of PM implementation in the region.

## 5. Conclusions

PM is perceived as a complex and expensive program. The study participants agreed that PM is of high priority and can improve healthcare delivery outcomes if given the attention and the identified challenges are addressed. The most cited challenges to implementing PM included a lack of specific policies and guidelines on PM; a lack of a supportive infrastructure and limited suppliers of equipment and consumables that are needed for a PM program; and financial issues due to other competing priorities in the health sector. Additional barriers include cultural, behavioral, and religious beliefs and less trained, motivated, and specialized healthcare providers in PM. This study provides insights on the status of PM implementation in Rwanda, and it provides baseline data as a starting point to equip the Rwandan healthcare system to use PM as a tool to improve the quality of healthcare in the country and the region.

### Key Recommendations


Advocacy to add a PM program into the current national health sector’s strategic plans, policies, and priorities.The MOH to provide a legal framework and revising existing policies and guidelines that specify the use of PM in the healthcare system in Rwanda.The University of Rwanda, in collaboration with the RBC, could plan to provide PM training opportunities for healthcare providers.The University of Rwanda, in collaboration with the MOH, could think about revising the existing undergraduate nursing and medical schools’ programs to include the concept of PM.Through the public–private partnership model (PPP), the government of Rwanda could support fund mobilization in order to equip health facilities with technologies and infrastructure that meet the required standards.Health sector leadership might consider the creation of specialized oncology centers in all or some referral hospitals with all needed genomic equipment and trained staff to serve the needs of the country and implement a PM program.Global scientists and experts in PM could think to generate a clinical protocol to guide PM treatment decisions in different healthcare settings.


## Figures and Tables

**Figure 1 jpm-13-01707-f001:**
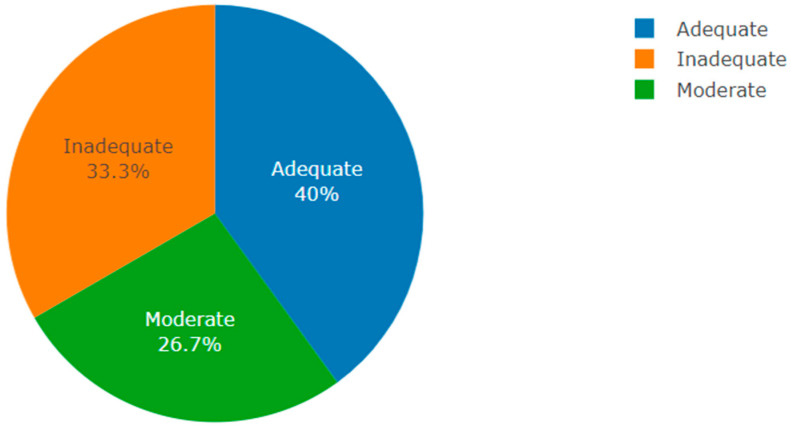
Perception of PM knowledge in Rwanda (author’s own figure).

**Figure 2 jpm-13-01707-f002:**
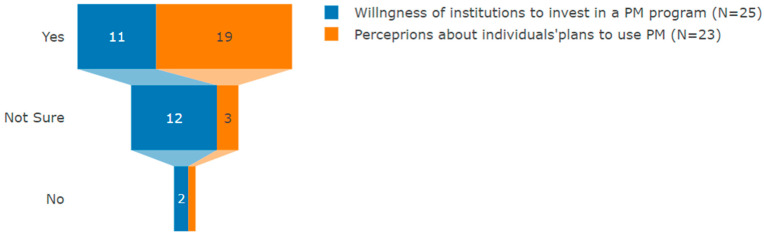
Willingness of institutions to invest in a PM program vs. participants’ perceptions of how they intend to use PM in their work (author’s own figure).

**Figure 3 jpm-13-01707-f003:**
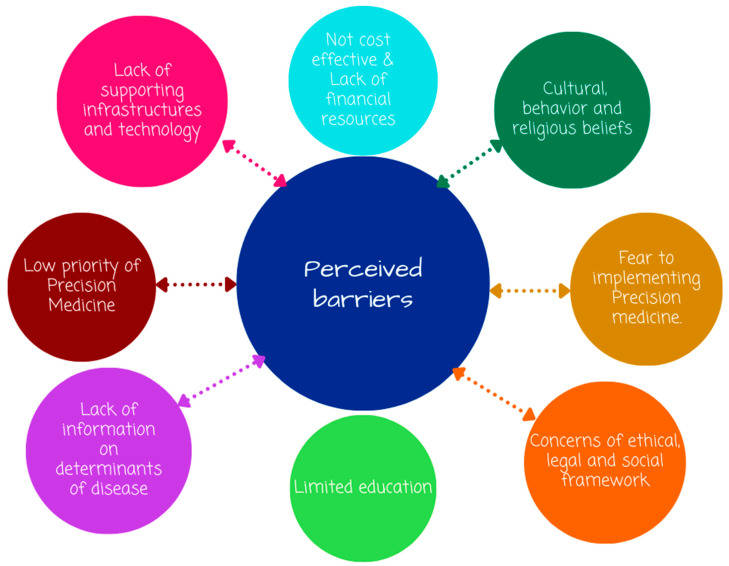
Perceived barriers to the implementation of precision medicine in Rwanda (author’s own figure).

**Figure 4 jpm-13-01707-f004:**
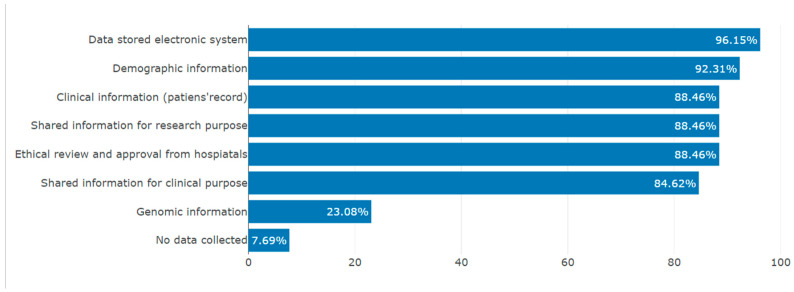
Routine patient data collection in institutions (author’s own figure).

**Table 1 jpm-13-01707-t001:** Proposed training needed for healthcare providers to enable implementation of precision medicine (PM).

Healthcare Providers’ Perceived Training Needs toward Precision Medicine	Answer	Frequency
Train laboratory technicians (N = 23)	Yes	21
	No	2
Train more genetic counsellors (N = 22)	Yes	22
	No	0
Create or expand a BS or MSc degree program (N = 22)	Yes	19
	No	3
Create a formal degree program (N = 22)	Yes	18
	No	4
Train our nurses and primary caregivers (N = 20)	Yes	19
	No	1
Send students and physicians abroad for specialized training (N = 19)	Yes	17
	No	2

## Data Availability

The original contributions presented in this study are included in this article. Further information can be obtained from the corresponding author upon reasonable request. Gratitude is extended to the National Council for Science and Technology (NCST) (grant number NCST-NRIF /ERG-BATCH1/P01/2019) for funding this project. We acknowledge the collaboration offered by the health professionals and decision makers who consented to participate in this study. Importantly, we thank the health sector leadership, mainly the Ministry of Health (MoH) and the Rwanda Biomedical Center (RBC). Special thanks is extended to the leaders from the hospitals that participated in this study—University Teaching Hospital of Kigali (CHUK), Butaro District Hospital (BDH), University Teaching Hospital of Butaro, Kigali Faisal Hospital (KFH), and Rwanda Military Hospital (RMH)—for their collaboration in the implementation of this study.

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
