# Peer review of "An Assessment of the Knowledge and Perceptions of Precision Medicine (PM) in the Rwandan Healthcare Setting"

_jpm, 2023, doi:10.3390/jpm13121707_

Round 1

Reviewer 1 Report

The subject matter of this article is important, not only in Rwanda, but throughout the Third World and even in the Western world. The conclusions are as expected and are very well reflected in the final part of the article. However, there are several problems that need to be highlighted:

a) The number of individuals on which the study is based, in each of its fields, is very small and therefore of doubtful statistical value. It is probable that in Rwanda the number of people cannot be extended but, then, the study or the different studies should be extended to other neighboring countries with the same problem.

b) With the data obtained, a first communication of a political or journalistic type could be elaborated, but without scientifically proven value.

For all these reasons, I would consider another version of the article, with the defects that I have described, and timely resolved with the previous suggestions. I suggest that this initiative continue, as a means of highlighting health problems that, once detected, may be in the process of being solved.

The level of the English is correct.

Author Response

Dear Reviewer, thank you for taking the time to review my article. Kindly consider the response to your important comments and inputs which were adjusted in the revised manuscript.

Reviewer 2 Report

1. Ethical approval reference number with date is required

2. The sentence flow is not right, kindly recheck and break in short sentence to understand clearly

Example

Fewer participants explained that the minimum to implement PM in Rwanda in the breast cancer cases can be done as the DNA can be extracted, the equipment for biobank, for investigation to identify cancers, for pathology, and for immuno-chemistry are available, but not to fully implement PM because the equipment to do the genome sequencing of the gene panel are not available and PM uses a lot of like molecular genetics

3.

The data collection and methodology following any standard procedure or author' s modified version or own process kindly mention.

4.

The following points advised to discuss

Precision medicine in metastatic breast cancer is the utilization of biomarkers,  liquid biopsy, precision radiotherapy as well as provision of individualized treatment to patients based on their genetic and molecular characteristics.

The cost of genetic testing and PM application also need to discuss.

 5.

Advised to cite the following articles as and where required

Subhan, M. A., Parveen, F., Shah, H., Yalamarty, S. S. K., Ataide, J. A., & Torchilin, V. P. (2023). Recent Advances with Precision Medicine Treatment for Breast Cancer including Triple-Negative Sub-Type. Cancers15(8), 2204. https://doi.org/10.3390/cancers15082204

Uwineza, ANduwayezu, RNgenzi, JLMahboob, UPrecision medicine education for medical residents in RwandaMed Educ202357(8): 775-776. doi:10.1111/medu.15100

https://twas.org/article/rwanda-rise

6. 

Source of Fig-- like authors own should be mentioned in parenthesis.

Author Response

(The authors gave the same response as above.)

Round 2

Reviewer 1 Report

The paper with the new modifications incorporated can be accepted for publication. From my point of view, the information you provide can be the starting point for procedures to be put into play that can improve precision medicine in Rwanda.

Reviewer 2 Report

The revisions made by the authors are satisfactory.